# Management Strategies and Nursing Activities for Nutritional Care in Hospitalized Patients with Cognitive Decline: A Scoping Review

**DOI:** 10.3390/nu14194036

**Published:** 2022-09-28

**Authors:** Gloria Liquori, Aurora De Leo, Daniele De Nuzzo, Victoria D’Inzeo, Rosario Marco Arancio, Emanuele Di Simone, Sara Dionisi, Noemi Giannetta, Francesco Ricciardi, Fabio Fabbian, Giovanni Battista Orsi, Marco Di Muzio, Christian Napoli

**Affiliations:** 1Department of Biomedicine and Prevention, Tor Vergata University of Rome, 00133 Rome, Italy; 2Nursing, Technical, Rehabilitation, Assistance and Research Direction—IRCCS Istituti Fisioterapici Ospitalieri—IFO, 00144 Rome, Italy; 3Department of Clinical and Molecular Medicine, Sapienza University of Rome, 00185 Rome, Italy; 4School of Nursing, UniCamillus—Saint Camillus International University of Health and Medical Sciences, 00131 Rome, Italy; 5Department of Medical Sciences, University of Ferrara, 44121 Ferrara, Italy; 6Department of Public Health and Infectious Diseases, Sapienza University of Rome, 00185 Roma, Italy; 7Department of Surgical and Medical Sciences and Translational Medicine, Sapienza University of Rome, 00185 Rome, Italy

**Keywords:** cognitive impairment, cognitive decline, nutritional care, hospitalized patient, elderly patient

## Abstract

Cognitive impairment and dementia can negatively impact the nutritional capacities of older people. Malnutrition is common in hospitalized frail elderly people with cognitive impairment and negatively affects prognosis. Malnutrition worsens the quality of life and increases morbidity and mortality. This scoping review aimed to identify factors affecting the risk of malnutrition and preventive strategies in hospitalized patients with cognitive impairment, focusing on nursing interventions. The authors researched population, context, and concept in international databases of nursing interest. Full texts that met the inclusion criteria were selected and reviewed. The extracted data were subject to thematic analysis. A five-stage approach, already reported in the scientific literature, was utilized in the following scoping review. Of 638 articles yielded, 9 were included. Two focus areas were identified as follows: (1) prevalence and risk factors of malnutrition in older patients with cognitive decline; (2) nursing strategies used to enhance clinical outcomes. Nursing health interventions aim to recognize and reduce malnutrition risk, positively impacting this phenomenon. A multidisciplinary team is essential to meet the nutritional needs of these patients.

## 1. Introduction

Due to the global increase in aging, the elderly population is constantly growing [1]. During the period 2015–2030, the elderly population is expected to grow from 901 million to 1.4 billion people (by 56%), and the 2015 population is expected to double by 2050 [2].

This increase in the elderly population is already placing substantial extra strain on healthcare and support services [3], increasing the costs due to health-related complications [4].

Among the main concerns linked to age, malnutrition is a common health problem in people older than 65 [5]; in fact, nutritional fragility is a frequent condition in vulnerable elderly people and is related to an increased incidence of mortality in this population [6]. Moreover, it is often associated with a reduced adaptive response to physiological and pathological conditions [7]; for instance, the elderly population experiences a physiological loss of taste, which impacts the frailty condition. Assessment of nutritional status through anthropometric measurements in these patients is essential to ensure healthy aging and adequate food intake [8].

Malnutrition has a negative impact on both people living independently and, overall, on those admitted to healthcare facilities, affecting up to 60% of hospitalized older adults [9]. In fact, according to estimates, up to 50% of patients are undernourished when they are admitted, increasing their malnutrition while being treated in a hospital [10]. This issue is linked to lengthier hospital stays, increased morbidity (pressure ulcers, infections, and falls), and mortality, especially in patients affected by chronic diseases [11,12].

In elderly people with cognitive impairment, the phenomenon is also more serious, since malnutrition irreversibly worsens other health conditions [13]. At the same time, mental status significantly affects nutritional status; people with lower cognitive levels tend to face a higher risk of malnutrition, especially during hospitalization [14,15]. The relationships between nutritional status, cognitive decline, and performance are complex and reciprocal: the presence or the risk of malnutrition may influence cognitive performance, and the presence of cognitive decline may affect the activities of daily living (ADL), also affecting food intake [16].

Unfortunately, malnutrition is also a frequently underdiagnosed entity, capable of subtly impacting patient outcomes, length of stay, hospital costs, and readmissions [9]. Recent studies suggest the crucial role of nurses in preventing, assessing, and treating malnutrition in this fragile population [2,7]. One of the possible key points could be implementing all known strategies to avoid worsening nutritional status, improving health status, and reducing mortality risk [17].

Therefore, this scoping review was targeted at evaluating the relationship between nursing activities and the identification, prevention, and management of malnutrition among hospitalized elderly individuals with cognitive impairment. In particular, our aims were collecting best practice and scientific evidence with regard to: (i) risk factors for developing low food intake in hospitalized older patients with cognitive impairment; (ii) prevalence of malnutrition in older patients with cognitive impairment; (iii) identify the nursing strategies to enhance clinical outcomes and care of patients with cognitive impairment in the hospital environments.

## 2. Materials and Methods

### 2.1. Literature Search

The five-step approach presented by Arksey and O’Malley [18], and advanced by Levac and collaborators [19] was utilized in the following scoping review. The foreseen steps were: 1. determining the study problem, 2. outlining relevant investigations, 3. studies selection, 4. data charting, 5. collating, summarization, and presenting the findings.

The choice of a scoping review was based on the need to identify the nature and extent of the research evidence in accordance with Grant et al. [20].

The study was conducted according to the Preferred Reporting Item for Systematic Review and Meta-analysis for Scoping Review (PRISMA-ScR) [21] (Appendix A).

### 2.2. Step 1: Determining the Study Problem

The objectives of the review were to provide answers to the following questions:What is the prevalence of malnutrition in older patients with cognitive impairment?What are the risk factors for developing low food intake in hospitalized older patients with cognitive impairment?Which nursing strategies are used to enhance clinical outcomes and patients with cognitive impairment care in the hospital environments?

The PCCT (population, concept, context, and type of study) methodology was utilized to identify search questions according to Peters et al. [22]. Specifically, the population included hospitalized elderly patients (aged > 65 years) with cognitive impairment and malnutrition. The concept was prevalence of phenomenon, nursing health interventions aimed at recognizing, reducing the risk of malnutrition, and positively affecting this phenomenon. The context was health care services admitting older people with cognitive impairment. With regards to type of study, all observational, experimental, and quasi-experimental studies with available full text in English, Spanish, and Italian were included.

### 2.3. Step 2: Outlining Relevant Investigations

A systematic search was conducted on the following scientific databases: PubMed, Cumulative Index to Nursing and Allied Health Literature (CINAHL), Psychological Abstracts Information Services (PsycINFO), Scopus, and National Library of Medicine (MEDLINE) via EBSCO and Cochrane Library. Observational, experimental, and quasi-experimental studies on malnourished elderly inpatients (>65 years) with cognitive impairment (including dementia and Alzheimer’s disease) in English, Spanish, and Italian were included. Studies involving people living at home and the adult and pediatric population (<65 years old) were excluded. No limits for country of origin or geographical context were applied. No time limits were applied.

### 2.4. Step 3: Study Selection

Citations were imported into Zotero^®^ Reference Manager, and the duplicates were eliminated. Two independent researchers conducted the initial screening, from March 2022 to May 2022, by reading the titles and abstracts of the publications. Unrelated studies were removed. If the publication’s relevance was undefined based on the title or abstract reading, the reviewers read the paper in full text to determine its eligibility.

The same investigators retrieved and assessed the whole text of articles deemed eligible for inclusion criteria. Any disagreement was resolved by discussion and final consensus. When the latter was not reached, arbitration was sought from a third researcher who supervised the study.

### 2.5. Step 4: Data Charting

The selected articles were summarized in Table 1 [23,24,25,26,27,28,29,30,31], including authors and year, aim, method, and main results.

### 2.6. Step 5: Collating, Summarization, and Presenting the Findings

Lastly, the results were collated and summarized according to Arksey and O’Malley’s framework [18], respecting the proposed search strategy.

## 3. Results

The search strategy yielded 638 articles; 142 duplicates were excluded. A further 421 records were excluded after applying the title and abstract eligibility criteria. The full texts of 75 articles were reviewed. Of these, nine articles met the inclusion criteria and were included in this scoping review [23,24,25,26,27,28,29,30,31]. Figure 1 shows the search and selection process according to the PRISMA statement [21]. (Figure 1).

The relationship between older people’s nutritional status, hospitalization, and cognitive impairment as a result of the included studies is shown in Table 1 and in the Appendix A Appendix A.

According to the scoping review framework, the main themes were divided into two results sections.

### 3.1. Prevalence and Risk Factors of Malnutrition in Older Patients with Cognitive Impairment

Orsitto et al. [23] described an extremely high prevalence of poor nutritional status in a sample of hospitalized older patients with different grades of cognitive impairment. Only 18% of the sample was well nourished, while 82% were at risk of malnutrition or malnourished. Findings showed a significantly greater malnutrition rate in hospitalized patients with severe cognitive impairment. This study showed the evidence of poor nutritional status even in patients with mild cognitive impairment who had not yet progressed to dementia [23]. Moreover, according to Salva et al. [28] patients with dementia showed a high risk of malnutrition with respect to other patients. According to Lin et al. [29], eating difficulty, no feeding assistance, moderate dependence, fewer family visits, and being female and older were six independent factors associated with low food intake after controlling for all other aspects [29].

Hospitalized frail patients develop a major risk of under-nutrition and weight loss [23,25]. However, according to the findings of another study [25] there are no differences in malnutrition among different groups of hospitalized patients concerning age, length of stay, gender, or baseline anthropometric scores.

### 3.2. Nursing Strategies Used to Enhance Clinical Outcomes

Simple, inexpensive, and easy-to-implement strategies, such as early dietary assessment; dietary “grazing” and staggered mealtimes, can improve nutrition in hospitalized elderly patients [25]. Nursing strategies that provide information on the clinical, functional, and cognitive aspects of the disease should be used in hospitalized patients, especially those with cognitive impairment [25]. The immediate evaluation of eating abilities, nutritional needs, and dietary preferences is a simple and inexpensive strategy that can lead to positive changes in nutritional intake in this population [25,31]. Indeed, assessing the patient’s nutritional needs early is critical to reducing hospitalization [24], as it improves the patient’s weight, but this does not affect cognitive impairment [25]. A positive element emerges from the study of Avelino-Silva et al.: hospitalization, by allowing more time to assess each patient, provides the opportunity for a detailed and structured nutritional clinical assessment through a CGA tool that has proven useful in reducing mortality in these patients [24]. Other tools have been used in order to evaluate nutritional status, for instance, Salva et al. used the mini nutritional assessment scale [28].

Some studies have also shown other strategies to enhance the nutritional outcome in patients with cognitive impairment. For example, the main objective of Lauque et al. [26] was to evaluate the effects of OS. Overall, 46 patients (intervention group) received 3-month OS, while the other 45 patients (control group) received standard care in geriatric wards and daycare centers in the Toulouse region. Protein and energy consumption considerably increased in the intervention group between baseline and 3 months, leading to a considerable gain in weight as well as fat-free mass. Nevertheless, no substantial changes in biological markers, cognitive function, or dependence were observed. Therefore, the authors conclude that the regular OS assumption can aid in preserving the gain in fat-free mass and enhance these individuals’ nutritional status. Additionally, a study conducted by Allen et al. [30] showed that supplement drinks may be beneficial in reducing the prevalence of malnutrition within the group, as more people meet their nutritional requirements. Moreover, another study by Allien et al. [27] showed that drinking nutritional beverages with a glass makes the patient more stimulated to drink rather than using a straw. Baumgartner et al. [31] found that individualized nutritional support improves functional outcomes and quality of life (QoL) over 30- and 180-day periods of nutritional support.

Finally, if nurses take the time to assess the nutritional status and needs, implement suitable care plans and provide food and drink in ways that ensure their safe consumption, this can positively affect both patients’ nutritional status and their general health condition, reducing the risk of mortality too [28,29,30,31].

## 4. Discussion

### 4.1. Nutritional Assessment and Screening

Malnutrition has a high prevalence in hospitalized elderly patients [25,26]. This review confirms that the potential associated factors are different: medical history, medicines intakes, diet, oral health, swallowing ability, physical and cognitive function, gastrointestinal, psychiatric, and neurological conditions, and also social aspects of a person’s life [24]. Therefore, every hospital should establish an interdisciplinary approach to nutrition care based on formal policies and procedures, ensuring the early identification of malnourished patients or malnutrition risk and implementing comprehensive nutrition care plans [24]. In fact, the review results suggest that patients should be screened for malnutrition within the first 24 h of admission and screened regularly during their hospital stay [23,24]. Moreover, it is critically important to establish individualized nutritional support to these patients, as this improves functional outcomes and QoL, as well as reducing mortality by 50% at 30 days in hospitalized elderly patients [31].

The MNA has become a tool allowed standardized, reproducible, and reliable determination of nutritional status [26,31]. However, some studies reported that the MNA-SF (mini nutritional assessment short-form) could overestimate malnutrition risk [28].

Avoiding and solving malnutrition in elderly individuals is a crucial element of geriatric care, since healthy people can also become malnourished during hospitalization. In this context, early dietary assessment and implementation of feeding strategies are crucial not only for vulnerable patients [25].

### 4.2. Nutritional Strategies and Management

Breaking the vicious circle between malnutrition and cognitive impairment can help patients and reduce the impact of this degenerative condition [25,28].

The management of elderly adults who are malnourished or at malnutrition risk should be multimodal and multidisciplinary, as reported by Salva et al. in their nutritional intervention program, “The NutriAlz”, aimed at preventing weight loss patients affected by dementia [28].

A routine and impersonalized hospital nutrition in elderly patients carries an increased risk of mortality compared with individualized nutritional support [31]. Especially, patients with dementia and cognitive impairment hospitalized for acute diseases often require individualized strategies to maintain adequate caloric intake.

Moreover, eating difficulties is a factor associated with low food intake [29]. Therefore, attention must be paid to oral frailty, defined as a gradual age-related loss of oral function along with decline in cognitive and physical function [32]. In addition, altered eating behavior and dysphagia are factors to be managed in these frail hospitalized patients [26]. These patients have impaired oral motor skills, particularly chewing function, oral diadochokinesis, swallowing and salivary disorders, all associated with few teeth left in the oral cavity [29,32].

### 4.3. Environmental Changes to Improve Mealtime Habitat and Experience

Environmental elements such as food accessibility (for example a glass door refrigerator with snack food easily visible), companions and furniture, smell, ambient sounds, lighting and temperature, size of the portion, eating location, and presentation of food play a crucial role during mealtimes, improving the patients’ compliance to eat [25]. It is demonstrated that changes in mealtime habits and atmosphere, based on the personal needs of patients, could increase nutritional intake and reduce the malnutrition risk [25,29], especially among older people.

A great variety of environmental variables might have an influence on the nutritional intake of elderly inpatients as also reported in the review findings [25,30]. Wong, et al. have shown that appetite increased if those people needing more assistance were fed earlier than other patients; it is not the time of meal initiation that is important, but the longer duration [25]. To reduce the prevalence of malnutrition in these patients, it is useful to offer supplementary drinks, as they can improve oral intake and increase appetite [30]. Another environmental change to improve mealtime habitats is music. Wong et al. have shown that patients spent more time at the table when music was played, so it appears to be an effective strategy to lengthen mealtime and increase patients’ appetite [25].

Family style meals, eating meals with caregivers, relaxing music during mealtime, patient education, protected mealtimes, and additional food assistance (implemented alone or in combination) were among the most promising interventions to improve mealtime experiences [25,29,30]. It is crucial that these fragile patients receive support from nurses or family members at mealtimes [25]. This relational strategy helps the patient to increase food intake during meals [29]. Receiving good mealtime assistance and increasing time spent by nurses or volunteers on feeding or helping during meals may positively affect eating behavior with a positive effect on the nutritional intake in older inpatients [27,29].

The physical presence of a caregiver helps the patient to be more focused on his or her meal [25,29]. Caregiver education is crucial to ensure proper weight maintenance for the patient who may go through weight loss even if they have a positive energy balance [26].

### 4.4. Limits of the Study

This scoping review focused only on the relationship between nursing activities and identification, prevention, and management of cognitive impairment in elderly hospitalized individuals, and the main limitation is the reduced availability of studies in the hospital environment. The majority of published studies recruited patients in the home setting. Hospitalization is usually due to further acute illnesses; therefore, general conditions and eventually chronic diseases associated with acute events worsen. Moreover, in this study, we could not take into consideration the intensity of care, including some invasive treatments such as intravenous or enteral support. Finally, we could not stratify nursing activities considering the cause of cognitive impairment, its worsening, its stage, or the duration of the cognitive impairment.

## 5. Conclusions

The findings of this scoping review suggest that malnutrition may have high frequency in hospitalized elderly patients, especially those affected by cognitive impairment. The relationship between nutritional status and cognitive impairment is complex and reciprocal. Therefore, an appropriate nutritional status evaluation along the hospitalization, followed by both healthcare and environmental managing strategies is necessary to maintain or improve the patients’ nutritional status. A multidisciplinary team is essential to fulfilling the nutritional needs of these patients, and the role of nursing activities is crucial.

## Figures and Tables

**Figure 1 nutrients-14-04036-f001:**
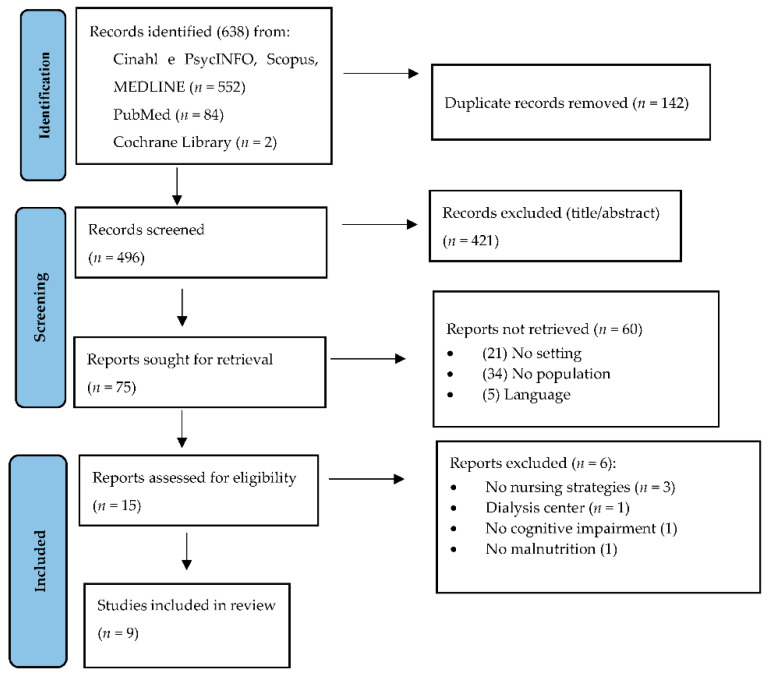
Flow diagram of the search and selection process, based on PRISMA flowchart.

**Table 1 nutrients-14-04036-t001:** Summary of findings.

Authors and Year	Aim	Method	Results
Lauque et al., 2004 [26]	To study the effects of OS (oral supplement) on body weight, body composition, nutritional status, and cognition in elderly patients with Alzheimer’s disease.	Prospective, randomized, controlled study. A total of 46 patients (intervention group) received 3-month OS. The other 45 patients (control group) received usual care.	Between baseline and 3 months, energy and protein intake significantly improved in the intervention group, resulting in a significant increase in weight and fat-free mass. No significant changes were found for dependence, cognitive function, or biological markers. The nutritional benefit was maintained in the intervention group after discontinuation of OS at 3 months.
Wong et al., 2008 [25]	To evaluate strategies designed to improve nutrition in elderly hospitalized patients with dementia.	Interventions: Phase 1: Observation. Phase 2: Encouraging dietary, “Grazing”. Phase 3: Using volunteers to feed patients. Phase 4: Improving dining room ambience by playing soothing music.	There were no differences between the groups concerning age, length of stay, gender, or baseline anthropometric scores. Simple, inexpensive and easy to implement strategies can improve nutrition in hospital in patients with dementia.
Orsitto et al., 2009 [23]	To assess the prevalence of malnutrition in older patients with mild cognitive impairment.	A total of 623 hospitalized elderly patients underwent the comprehensive geriatric assessment to evaluate medical, cognitive, affective and social aspects. Nutritional status was assessed by using the mini nutritional assessment. According to the neuropsychological evaluation cognitive function was categorized into three levels—normal cognition, mild cognitive impairment and dementia.	According to the mini nutritional assessment classification, 18% of the sample study was assessed as well-nourished, 58% at risk of malnutrition and 24% as malnourished. Patients with mild cognitive impairment and dementia had a significantly lower frequency of well-nourished and higher frequency of being at risk of malnutrition or malnourished than patients with normal cognition.
Salva et al., 2009 [28]	To describe the study design, intervention program, recruitment, randomization, and patients’ baseline characteristics.	Intervention: the NutriAlz program A personalized presentation and handover of a briefcase. Training for families, caregivers. Support in weight monitoring. Periodic information for the families. Action protocols and standardized help decision.	Evaluation of the risk for malnutrition using the mini nutritional assessment resulted in 5% of malnourished patients, 37% at risk for malnutrition and 58% well-nourished subjects. The MNA score was significantly different between the two groups. Patients with dementia showed a high risk of malnutrition.
Lin et al., 2010 [29]	To investigate the risk factors of older people with dementia for developing low food intake.	Four hundred seventy-seven participants with dementia from nine dementia special care units in licensed long-term care facilities (LTCFs) in Northern and Central Taiwan. Data were collected using the Barthel index, Mini Mental State Examination, and Edinburgh Feeding Evaluation in Dementia scale.	The prevalence of low food intake at meals in patients with dementia in LTCFs was 30.7%. Eating difficulty, no feeding assistance, moderate dependence, fewer family visits, and being female and older were six independent factors associated with low food intake after controlling for all other aspects.
Allen et al., 2013 [30]	Investigate the impact of the provision of OS on protein and energy intake from food and the ability to meet protein and calorie requirements in people with dementia.	After consent by proxy was obtained, participants were enrolled in a cross-over study comparing oral intake on an intervention day to an adjacent control day.	More people achieved their energy and protein requirements with the supplement drink intervention without sufficient impact on habitual food consumption. Findings from these 26 participants with dementia indicate that supplement drinks may be beneficial in reducing the prevalence of malnutrition within the group as more people meet their nutritional requirements. As the provision of supplement drinks has an additive effect on consumption of habitual foods, these can be used alongside other measures to also improve oral intake.
Allen et al., 2014 [27]	To analyze the influence of the serving method on compliance and consumption of nutritional supplement drinks in older adults with cognitive impairment.	Participants were randomized to the serving method. Nursing and care staff were instructed to give the supplement drinks three times per day on alternate days over a week by the allocated serving method. The researcher weighed the amount of supplement drink remaining after consumption.	Participants randomized to consume nutritional drinks from a glass/beaker drank statistically significantly more than those who consumed them via a straw inserted directly into the container. However, supplements placed in a glass/beaker were more frequently omitted. Nutritional supplement drinks should be given to people with dementia who are able to feed themselves in a glass or a beaker if staffing resources allow.
Avelino-Silva et al., 2014 [24]	Assess the applicability of the proposed model comprehensive geriatric assessment (CGA) for thoroughly characterizing patients with cognitive impairment and analyze the impact of this strategy on the prediction of mortality and adverse hospital outcomes.	This prospective observational study included 746 patients aged 60 years and over. The proposed CGA was applied to evaluate all patients at admission. Impairment in ten CGA components was mainly investigated: polypharmacy, activities of daily living (ADL) dependency, instrumental activities of daily living (IADL) dependency, depression, dementia, delirium, urinary incontinence, falls, malnutrition, and poor social support.	CGA was a useful tool to identify patients at higher risk of in-hospital death and adverse outcomes, of which those with malnutrition were foremost.
Baumgartner et al., 2021 [31]	This article aimed to study the effects of individualized nutritional support for patients with ageing-related vulnerability in the acute hospital setting on mortality and other clinical outcomes.	The study analyzed data of patients at nutritional risk (Nutritional Risk Screening 2002 score ≥ 3 points) with ageing-related vulnerability, randomized to receive protocol-guided individualized nutritional support to reach specific protein and energy goals (intervention group) or routine hospital food (control group). The primary endpoint was all-cause 30 d mortality. Trained study nurses performed structured telephone interviews with all patients 30 days after inclusion to collect outcome information.	This study found a more than 50% reduction in mortality at 30 days in hospitalized patients with ageing-related vulnerability at nutritional risk receiving protocol-guided individualized nutritional support to reach specific protein and energy goals. Significant improvements were also found for longer-term mortality at 180 days. Individualized nutritional support also improved functional outcomes and quality of life (QoL) over 30 and 180 days. These data support the early screening of hospitalized patients with aging-related vulnerability for nutritional risk, followed by the implementation of individualized nutritional interventions.

## Data Availability

All data are available upon request.

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
