# Peer review of "Management Strategies and Nursing Activities for Nutritional Care in Hospitalized Patients with Cognitive Decline: A Scoping Review"

_nutrients, 2022, doi:10.3390/nu14194036_

Round 1
Reviewer 1 Report
Thank you for involving me in the review. Some comments below:
Introduction: In line 7, it is appropriate to expand the concept of malnutrition and elaborate on the concept of nutritional frailty, a typical condition of malnutrition in the elderly (10.1016/j.arr.2020.101148 and 10.1111/joim.13384).
Methodology: Line 102. it is appropriate to tabulate the search strategy according to PECO/PICO guidelines (depending on the type of studies the authors chose to include) instead of pasting the QUERY from Pubmed.
Line 146: In Table 3, please reduce the content of the sections, and delete the TITLE column. It is difficult to read.
Line 225 of the discussions, "Altered eating behavior, dysphagia, impaired communication, agitation, and decreased level of consciousness are factors to manage in these frail hospitalized patients" please expand the concept of oral frailty (10.1016/S2666-7568(21)00143-4).
Author Response
Dear Reviewer,
We highly appreciate you detailed and valuable comments on our manuscript “Management Strategies and Nursing Activities for Nutritional Care in Hospitalized Patients with Cognitive Decline: a Scoping Review”.
The suggestions were greatly helpful for us, and we addressed all comments in the revised paper. By referring also to the new suggested references, we have almost rewritten the manuscript to improve its quality. We did our best to revise it and we hope these efforts will be worked. On behalf of my co-authors, I would like to clarify point by point the concerns raised:
- Reviewer’s comments – Introduction: In line 7, it is appropriate to expand the concept of malnutrition and elaborate on the concept of nutritional frailty, a typical condition of malnutrition in the elderly (10.1016/j.arr.2020.101148 and 10.1111/joim.13384).
Answer: Done! Thank you for your kind review. We added new sentences to expand the concept of malnutrition and frailty. The suggested new references were added in the reference list and discussed.
- Reviewer’s comments –Methodology: Line 102. it is appropriate to tabulate the search strategy according to PECO/PICO guidelines (depending on the type of studies the authors chose to include) instead of pasting the QUERY from Pubmed.
Answer: Thank you for your comment. We identified the search questions according to PCCT methodology in accordance with Grant et al. and Arksey and O'Malley. We better explained this method in the appropriate section. QUERY from Pubmed was removed.
- Reviewer’s comments – Line 146: In Table 3, please reduce the content of the sections, and delete the TITLE column. It is difficult to read.
Answer: Done! Thank you for your comment.
- Reviewer’s comments – Line 225 of the discussions, "Altered eating behavior, dysphagia, impaired communication, agitation, and decreased level of consciousness are factors to manage in these frail hospitalized patients" please expand the concept of oral frailty (10.1016/S2666-7568(21)00143-4).
Answer: Done! Thank you. The text has been improved accordingly. We have expanded the concept of oral frailty in the discussions referring to the suggested new reference, that was added in the reference list.
THANK YOU FOR YOUR TIME

Reviewer 2 Report
I hope you read my comments as constructive criticism rather than being negative.
Abstract, line 23. I would suggest saying ‘can negatively impact’, as otherwise you are saying that all cases of cognitive impairment and dementia have a negative impact, and I’d argue that’s not necessarily the case. There are different stages/levels of cognitive impairment and dementia, so implying that nutritional capacity is affected for everyone with one of the conditions is inaccurate. (in line 61 you say ‘may affect’ so why not do the same here?)
Abstract, line 24. ‘frail elderly people’ is preferable, as I find ‘frail elderly’ to be dehumanising
Lines 49-50. ‘It impacts people living independently and those in hospital’ might be better and easier to understand.
Line 56. ‘malnutrition worsens irreversibly the general conditions’ does not read very well. Something like ‘malnutrition irreversibly worsens other conditions’ or ‘malnutrition worsens other conditions irreversibly’ might be better
Line 59. ‘sensibly rises’ does not makes sense. I’m not actually sure what you mean. Please consider rewording this.
Line 63. I think you might mean ‘capable’ rather than ‘able’ – it doesn’t read very well at the moment
Lines 67-70. You seem to be using ‘dementia’ and ‘cognitive impairment’ and ‘mild cognitive impairment’ almost interchangeably. They are not the same thing, so I think you need to be consistent and clearer about which group of patients you are actually looking at.
Line 76. I think ‘five steps’ would be better than ‘five-stage steps’
Lines 76-78 and the subheadings on lines 83, 93, 103, 113 and 115. I would be consistent with the language you use. The first step you specify in line 76 should be the same as in line 83, rather than slightly different ways of saying the same thing. I.e. use either ‘determining the study problem’ or ‘identifying the research question’, rather than both.
I think you need to be consistent with your tenses, which I suggest should be the past tense as you are talking about what you did, not what you are doing. For example, in line 65 ‘This scoping review was targeted’, line 67 ‘Our aims were collecting’, line 79 ‘review was based’. In line 75 you say ‘was utilized’, which is what I would expect to see.
Lines 84-91. I think there needs to be more explanation about how the five questions relate to table 1, as it is not clear at present. Also, in line 92 you don’t say what PCCT means. I think there is more work to be done around this section.
Line 100. Unless you specifically mean patients, I would suggest saying ‘people living at home’ rather than ‘home patients’
Line 114. I’d suggest adding in something to say what information was summarised in the table, such as aim, method, results (basically the column titles from Table 3).
Line 121. You say that 76 articles were reviewed, but in the PRISMA diagram, line 134, you say 75. Which is correct?
Line 121. You say that 9 articles were included, but in the abstract (line 31) you say ten. Which is correct?
Table 3. I appreciate this may be the style used by the journal, but I find it quite difficult to read this table as the gap between columns is not very big. In some cases, it is not clear where the sentence in one column wraps round, especially when there is a lot of text.
Table 3. Sylvie Lauque article. Why do you say Sylvie Lauque and not just Lauque like you’ve got in the references?
Table 3, the Li-Chan Lin article. Why do you say Li-Chan Lin and not just Lin like you’ve got in the references?
Table 3, the second Allen et al. article. Should the year be 2014 as that’s what it says in the references [22]. Which is correct?
Table 3, the Avelino-Silva article. In the final column, should it be CGA instead of GCA?
I also find it odd that for the above article you expand some of the abbreviations, but in the rest of the table you don’t, but rely on a key given after the table. I would use one way or the other, rather than a mix
Line 199. I’d suggest removing ‘great’ as I don’t think it is necessary or particularly appropriate.
Line 219. I’d suggest using ‘degenerative’ or ‘progressive’ instead of ‘deadly’.
Lines 268-269. Please can you find an alternative to ‘admitted elderly subjects’ that is less dehumanising? Something like ‘elderly people admitted to hospital’ maybe?
Line 274. I’d suggest ‘can be very complex’ rather than ‘are very complex’. You are currently implying that all elderly patients are very complex, which is unlikely to be the case.
I’m struggling with the article, and I can’t quite work out why. I think it’s because the results give a very brief summary of the 9 articles, then the discussion pretty much ignores those 9 articles and brings in lots of additional references to make the key points around assessment, strategies and environment. In the abstract you imply that those three areas came from the scoping review articles, but it feels like that link isn’t really there in the article itself. Maybe if somewhere in the results/discussion those three areas were brought out through sub-headings and the relevant scoping articles given within each area before bringing in additional articles it might be clearer. At the moment I kind of feel a bit like ‘why did you bother doing the scoping review when you’ve already got all these other articles that make the points for you?’ Sorry, I realise that’s probably not very well explained.
Minor points
Line 44. I’d delete ‘also’ as I don’t think it’s necessary
Line 45. I’d delete ‘also’ as I don’t think it’s necessary
Author Response
Dear Reviewer,
We highly appreciate you detailed and valuable comments on our manuscript “Management Strategies and Nursing Activities for Nutritional Care in Hospitalized Patients with Cognitive Decline: a Scoping Review”.
The suggestions were greatly helpful for us, and we addressed all comments in the revised paper. As suggested, by removing unnecessary references from the discussion, we have almost rewritten the manuscript to improve its quality. We did our best to revise it and we hope these efforts will be worked. On behalf of my co-authors, I would like to clarify point by point the concerns raised:
- Reviewer’s comments – Abstract, line 23. I would suggest saying ‘can negatively impact’, as otherwise you are saying that all cases of cognitive impairment and dementia have a negative impact, and I’d argue that’s not necessarily the case. There are different stages/levels of cognitive impairment and dementia, so implying that nutritional capacity is affected for everyone with one of the conditions is inaccurate. (in line 61 you say ‘may affect’ so why not do the same here?)
Answer: Done! Thanks for your comment. The text has been changed accordingly.
- Reviewer’s comments – Abstract, line 24. ‘frail elderly people’ is preferable, as I find ‘frail elderly’ to be dehumanizing.
Answer: Done. Thank you. The text has been changed accordingly.
- Reviewer’s comments – Lines 49-50. ‘It impacts people living independently and those in hospital’ might be better and easier to understand.
Answer: Done. The text has been changed, improving the clearness. The text was changed as follow:
“Malnutrition has a negative impact on both people living independently and, overall, on those admitted to healthcare facilities, affecting up to 60% of hospitalized older adults”
- Reviewer’s comments – Line 56. ‘malnutrition worsens irreversibly the general conditions’ does not read very well. Something like ‘malnutrition irreversibly worsens other conditions’ or ‘malnutrition worsens other conditions irreversibly’ might be better
Answer: Thank you for your suggestions. The text has been changed accordingly. The text was changed as follow: “malnutrition irreversibly worsens other health conditions”
- Reviewer’s comments - Line 59. ‘sensibly rises’ does not makes sense. I’m not actually sure what you mean. Please consider rewording this.
Answer: Great! Thank you for your kind review. We removed the sentence and deepen the concept of the reciprocal relationships between nutritional status and cognitive decline.
- Reviewer’s comments Line 63. I think you might mean ‘capable’ rather than ‘able’ – it doesn’t read very well at the moment
Answer: Done! Thank you for your kind review.
- Reviewer’s comments – Lines 67-70. You seem to be using ‘dementia’ and ‘cognitive impairment’ and ‘mild cognitive impairment’ almost interchangeably. They are not the same thing, so I think you need to be consistent and clearer about which group of patients you are actually looking at.
Answer: Thank you for your kind review. We are interested in patient with cognitive impairment, including patients with dementia, and Alzheimer’s disease also. The term “impairment” was uniformed along the text and the definition was better explained in the methods section.
- Reviewer’s comments – Line 76. I think ‘five steps’ would be better than ‘five-stage steps’
Answer: The text has been changed accordingly.
- Reviewer’s comments – Lines 76-78 and the subheadings on lines 83, 93, 103, 113 and 115. I would be consistent with the language you use. The first step you specify in line 76 should be the same as in line 83, rather than slightly different ways of saying the same thing. I.e. use either ‘determining the study problem’ or ‘identifying the research question’, rather than both.
Answer: Done. The text has been changed accordingly.
- Reviewer’s comments – I think you need to be consistent with your tenses, which I suggest should be the past tense as you are talking about what you did, not what you are doing. For example, in line 65 ‘This scoping review was targeted’, line 67 ‘Our aims were collecting’, line 79 ‘review was based’. In line 75 you say ‘was utilized’, which is what I would expect to see.
Answer: Thank you for your suggestion. The text has been changed accordingly.
- Reviewer’s comments – Lines 84-91. I think there needs to be more explanation about how the five questions relate to table 1, as it is not clear at present. Also, in line 92 you don’t say what PCCT means. I think there is more work to be done around this section.
Answer: Done! Thank you for your suggestion. In the text has been deepened PCCT framework and we have explained questions relate to table 1. We added: “PCCT methodology (Population, Concept, Context and Type of study) was utilized to identify search questions according to JBI Reviewer’s manual Ch. 11 [21]. This framework helped us to check whether the inclusion ad the exclusion criteria were met or not”.
- Reviewer’s comments – Line 100. Unless you specifically mean patients, I would suggest saying ‘people living at home’ rather than ‘home patients’
Answer: Done! The text has been changed accordingly.
- Reviewer’s comments –. Line 114. I’d suggest adding in something to say what information was summarised in the table, such as aim, method, results (basically the column titles from Table 3).
Answer: Done! Thank you for your kind review. We added in the text: “…including aim, method and main results”.
- Reviewer’s comments - Line 121. You say that 76 articles were reviewed, but in the PRISMA diagram, line 134, you say 75. Which is correct?
Answer: Thanks for your comment. The text has been corrected. 75 articles were reviewed.
- Reviewer’s comments – Line 121. You say that 9 articles were included, but in the abstract (line 31) you say ten. Which is correct?
Answer: Done! Thank you for your kind review. The text has been corrected. 9 articles were included in the review.
- Reviewer’s comments – Table 3. I appreciate this may be the style used by the journal, but I find it quite difficult to read this table as the gap between columns is not very big. In some cases, it is not clear where the sentence in one column wraps round, especially when there is a lot of text.
Answer: Thank you for your comment. The text in the table has been changed.
- Reviewer’s comments – Table 3. Sylvie Lauque article. Why do you say Sylvie Lauque and not just Lauque like you’ve got in the references?
Answer: Done! Thank you. The text has been changed accordingly.
- Reviewer’s comments – Table 3, the Li-Chan Lin article. Why do you say Li-Chan Lin and not just Lin like you’ve got in the references?
Answer: Thank you! The text has been changed accordingly.
- Reviewer’s comments – Table 3, the second Allen et al. article. Should the year be 2014 as that’s what it says in the references [22]. Which is correct?
Answer: Thanks for your kind review. The text has been corrected. The second Allen et al. was written in 2014.
- Reviewer’s comments –. Table 3, the Avelino-Silva article. In the final column, should it be CGA instead of GCA?
Answer: Thanks for your kind review. The text has been corrected. The right term is CGA.
- Reviewer’s comments - I also find it odd that for the above article you expand some of the abbreviations, but in the rest of the table you don’t, but rely on a key given after the table. I would use one way or the other, rather than a mix.
Answer: Great! Thank you for your kind review. The key after the table has been removed, so the abbreviations has been expanded in the table text.
- Reviewer’s comments – Line 199. I’d suggest removing ‘great’ as I don’t think it is necessary or particularly appropriate.
Answer: Thank you for your kind review. The text has been changed accordingly.
- Reviewer’s comments – Line 219. I’d suggest using ‘degenerative’ or ‘progressive’ instead of ‘deadly’.
Answer: Done! The text has been changed accordingly.
- Reviewer’s comments - Lines 268-269. Please can you find an alternative to ‘admitted elderly subjects’ that is less dehumanising? Something like ‘elderly people admitted to hospital’ maybe?
Answer: Thank you for your suggestion. The text has been changed accordingly. Moreover, we removed the term “subjects” along the text.
- Reviewer’s comments – Line 274. I’d suggest ‘can be very complex’ rather than ‘are very complex’. You are currently implying that all elderly patients are very complex, which is unlikely to be the case.
Answer: The text has been completely changed in order to better underline the conclusions.
- Reviewer’s comments – I’m struggling with the article, and I can’t quite work out why. I think it’s because the results give a very brief summary of the 9 articles, then the discussion pretty much ignores those 9 articles and brings in lots of additional references to make the key points around assessment, strategies and environment. In the abstract you imply that those three areas came from the scoping review articles, but it feels like that link isn’t really there in the article itself. Maybe if somewhere in the results/discussion those three areas were brought out through sub-headings and the relevant scoping articles given within each area before bringing in additional articles it might be clearer. At the moment I kind of feel a bit like ‘why did you bother doing the scoping review when you’ve already got all these other articles that make the points for you?’ Sorry, I realise that’s probably not very well explained.
Answer: Thank you for this crucial comment. We completely agree with your suggestion and, for this reason, the results section was improved by detailing our findings and the discussion was deepened based on the review results and all the unnecessary references were removed from the discussion. The key points (assessment, strategies and environment) were also highlighted through specific subheadings.
- Reviewer’s comments – Line 44. I’d delete ‘also’ as I don’t think it’s necessary
Answer: The text has been changed accordingly.
- Reviewer’s comments – Line 45. I’d delete ‘also’ as I don’t think it’s necessary
Answer: The text has been changed accordingly.
THANK YOU FOR YOUR TIME
